# Single-Cell Molecular Characterization to Partition the Human Glioblastoma Tumor Microenvironment Genetic Background

**DOI:** 10.3390/cells11071127

**Published:** 2022-03-26

**Authors:** Francesca Lessi, Sara Franceschi, Mariangela Morelli, Michele Menicagli, Francesco Pasqualetti, Orazio Santonocito, Carlo Gambacciani, Francesco Pieri, Filippo Aquila, Paolo Aretini, Chiara Maria Mazzanti

**Affiliations:** 1Section of Genomics and Transcriptomics, Fondazione Pisana per la Scienza, San Giuliano Terme, 56017 Pisa, Italy; s.franceschi@fpscience.it (S.F.); m.morelli@fpscience.it (M.M.); m.menicagli@fpscience.it (M.M.); p.aretini@fpscience.it (P.A.); c.mazzanti@fpscience.it (C.M.M.); 2Department of Radiation Oncology, Azienda Ospedaliera Universitaria Pisana, University of Pisa, 56126 Pisa, Italy; francep24@hotmail.com; 3Department of Oncology, University of Oxford, Oxford OX1 2JD, UK; 4Division of Neurosurgery, Spedali Riuniti di Livorno—USL Toscana Nord-Ovest, 57124 Livorno, Italy; orazio.santonocito@uslnordovest.toscana.it (O.S.); carlo.gambacciani@uslnordovest.toscana.it (C.G.); francesco.pieri@uslnordovest.toscana.it (F.P.); filippo.aquila@uslnordovest.toscana.it (F.A.)

**Keywords:** single-cell, glioblastoma, tumor microenvironment, copy number aberrations, DEPArray

## Abstract

Background: Glioblastoma (GB) is a devastating primary brain malignancy. The recurrence of GB is inevitable despite the standard treatment of surgery, chemotherapy, and radiation, and the median survival is limited to around 15 months. The barriers to treatment include the complex interactions among the different cellular components inhabiting the tumor microenvironment. The complex heterogeneous nature of GB cells is helped by the local inflammatory tumor microenvironment, which mostly induces tumor aggressiveness and drug resistance. Methods: By using fluorescent multiple labeling and a DEPArray cell separator, we recovered several single cells or groups of single cells from populations of different origins from IDH-WT GB samples. From each GB sample, we collected astrocytes-like (GFAP+), microglia-like (IBA1+), stem-like cells (CD133+), and endothelial-like cells (CD105+) and performed Copy Number Aberration (CNA) analysis with a low sequencing depth. The same tumors were subjected to a bulk CNA analysis. Results: The tumor partition in its single components allowed single-cell molecular subtyping which revealed new aspects of the GB altered genetic background. Conclusions: Nowadays, single-cell approaches are leading to a new understanding of GB physiology and disease. Moreover, single-cell CNAs resource will permit new insights into genome heterogeneity, mutational processes, and clonal evolution in malignant tissues.

## 1. Introduction

Glioblastoma (GB) is the most aggressive and deadly primary tumor of the central nervous system in adults with an overall survival of fewer than 15 months [1]. The extremely poor prognosis of GB, despite the development in recent decades of new and innovative therapies, is enhanced by the resistance developed towards radio and chemotherapy [2]. In this tumor, as well as in other cancer types, the tumor microenvironment (TME) plays a pivotal role in treatment resistance [2]. The GB microenvironment is composed of a massive number of different cells, and besides malignant astrocytes and cancer stem cells, stromal cells, endothelial cells, pericytes, and a huge number of immune cells are present [3]. Moreover, intratumoral heterogeneity (ITH), which is one of the major features of GB tumors, is also hugely involved in anticancer treatment resistance [4,5] and is critical to promote tumoral growth and aggressiveness [6]. In support of this last remark, it has recently been demonstrated in GB that different sub-clones co-exist within the same tumor that respond differently to differing therapies [7]. These sub-populations of cells show distinct genomic profiles that reveal an individual behavior peculiar to the whole cell population [8]. Currently, the single-cell approach in GB is becoming increasingly popular. Reaching single-cell resolution enables avoiding the averaging of bulk analysis and capturing the heterogeneity of cells. Copy number aberration (CNA) is one of the most important somatic alterations in cancer [9,10], defined as somatic changes to the chromosome structure such as the gain and/or deletion of a particular DNA segment (>1 kb) [11]. The most common CNAs in GB include the loss, or partial loss, of chromosomes 9 and 10; the gain of chromosomes 7, 19, and 20; the focal deletion of the CDKN2A/B locus (9p21.3); and the focal high-level amplification of the EGFR locus (7p11.2) [12,13]. In particular, it is well known that CNAs targeting chromosomes 7 and 10 are some of the earliest events in GB tumor evolution [14]. The analysis of these aberrations is interesting because CNAs are detected with much greater accuracy than individual mutations and are associated with ITH in most cancers [15]. Moreover, the aggregation of cells sharing the same CNA profiles allows improving the phylogenetic analysis at the single nucleotide level [16].

In this work, we collected three human GB tumors and after dissociation, a certain number of single and groups of single cells were isolated through DEPArray technology, paying particular attention to four cell populations: astrocytes-like, microglia-like cells, endothelial-like cells, and stem-like cells. Afterwards, we investigated the genomic aberrations (CNA analysis) in these different types of tumor cells, thus performing a single-cell CNA analysis. The whole parental tumors were subjected to a bulk CNA analysis as well, to compare their molecular profiles with the single-cell results. The tumor partition in its single components allowed single-cell molecular profiling which revealed new aspects of the GB altered genetic background. Our work demonstrates that the single-cell approach is more representative and detailed than the bulk analysis, which contributes to a deeper insight into the basic molecular mechanisms of GB. Moreover, we presented an innovative approach to isolate and characterize different tumor populations of cells at the single-cell level.

## 2. Materials and Methods

### 2.1. Human Glioblastoma Tissue Collection

The study has been performed according to the Declaration of Helsinki and the samples’ collection protocol was approved by the Ethics Committee of the University Hospital of Pisa (787/2015). Tumor tissues were obtained from patients who underwent surgical resection of histologically confirmed GB after informed consent. Samples were obtained from the Unit of Neurosurgery of Livorno Civil Hospital. Three patient cases (GB01, GB02, and GB03) were included in the present study, the clinical and demographic data and the pathological and therapeutical information are summarized in Table 1. All cases had a diagnosis of GB with no previous history of any brain neoplasia and were not carrying R132 IDH1 or R172 IDH2 mutations. Surgically resected tumors were collected and stored in MACS tissue storage solution (Miltenyi Biotec, Bergisch Gladbach, Germany) at 4 °C for 2–4 h. Each tumor sample was washed with Dulbecco’s phosphate-buffered saline (DPBS) in a sterile dish and portioned with a scalpel into about 0.5–2 cm^2^ pieces under a biological hood. Afterward, they were vital frozen at −140 °C in 90% fetal bovine serum (FBS) and 1% dimethyl sulfoxide (DMSO) for further analyses.

### 2.2. Tumor Dissociation to Single-Cell Suspensions

Frozen GB tissues were defrosted in a water bath at 37 °C, washed with DPBS in a sterile dish and cut with a scalpel into small pieces. We used 0.11 gr, 0.16 gr, and 0.14 gr of GB01, GB02, and GB03 respectively. These finely minced tumor chunks were transferred in a C-tube (Miltenyi Biotec, Bergisch Gladbach, Germany) with the appropriate volume of buffer X following the protocol (Brain Tumor Dissociation Kit, Miltenyi Biotec, Bergisch Gladbach, Germany) for tumor dissociation with the gentleMACs Dissociator (Miltenyi Biotec, Bergisch Gladbach, Germany).

### 2.3. Immunofluorescence of Single-Cell Suspensions

The cell suspensions obtained were transferred to 1.5 mL LoBind tubes and washed three times with DPBS. After centrifugation at 300× *g* for 10 min at room temperature, the supernatant was removed, and the cells were resuspended in 400 µL of running buffer composed of MACS BSA stock solution (Miltenyi Biotec, Bergisch Gladbach, Germany) 1:20 with autoMACS Rinsing Solution (Miltenyi Biotec, Bergisch Gladbach, Germany). The cells were fixed by adding 400 µL of paraformaldehyde 4%; cells were incubated with fixation solution for 20 min at room temperature. To stop the reaction, the sample tubes were filled with DPBS and centrifuged at 400× *g* for 5 min at room temperature. Afterward, we performed two washes with DPBS to the sample tubes and then we incubated the pellet with blocking solution for 10 min at room temperature (BSA 3% in DPBS). The blocking reaction was stopped by filling the tube with DPBS, before centrifugation at 400× *g* for 5 min at room temperature. The cells were resuspended in running buffer and counted with a Luna Automated Cell Counter (Logos Biosystems, Anyang-si, Gyeonggi-do, Korea). For the immunofluorescence, a maximum of 100,000 fixed cells was used for the staining. The antibodies chosen for the staining were: anti-GFAP APC (130-124-040, Miltenyi Biotec, Bergisch Gladbach, Germany) for astrocytes, anti-IBA1 PE (ab209942, Abcam, Cambridge, UK) for microglia/macrophages cells, anti-CD105 PerCP/Cy5.5 (ab234265, Abcam, Cambridge, UK) for endothelial cells, anti-CD133 FITC (11-1339-42, eBioscience, San Diego, CA, USA) for stem cells, and Hoechst 33342 (62249, Thermofisher Scientific, Waltham, MA, USA) for nuclei. A total of 20 µL of anti-CD105 and 25 µL of anti-CD133 were added to the cell suspensions and mixed by gently pipetting. The samples were incubated for 15 min in the dark at 4 °C. The reaction was stopped by adding 1 mL of running buffer and mixed by gently pipetting. Then, the sample tubes were centrifuged at 400× *g* for 10 min at room temperature, the supernatant was removed, and the cells were resuspended with 100 µL of Inside Perm Buffer (Inside Stain Kit, Miltenyi Biotec, Bergisch Gladbach, Germany). A total of 8 µL of anti-GFAP and 2.5 µL of anti-IBA1 were added to the cell suspensions and mixed by gently pipetting. The samples were incubated for 20 min in the dark at room temperature. The reaction was stopped by adding 1 mL of Inside Perm Buffer, and mixed by gently pipetting. Then, the sample tubes were centrifuged at 400× *g* for 10 min at room temperature, the supernatant was removed, and the cells were resuspended with 1 mL of running buffer. Then, 1 µL of Hoechst (1 mg/mL) was added to the sample tubes and mixed by gently pipetting. The samples were incubated for 5 min in the dark at room temperature. Then, the sample tubes were centrifuged at 400× *g* for 10 min at room temperature and resuspended in 200 µL of running buffer.

### 2.4. Single-Cell Isolation by DEPArray^TM^ NxT

Single cells were isolated and sorted with DEPArray NxT (Menarini, Silicon Biosystems, Bologna, Italy). After the immunofluorescence of the single cell suspensions was measured, the cells were counted; we used a maximum of 24,000 cells to load the DEPArray NxT Cartridge. The samples were washed two times with 1 mL of SB115 Buffer (Menarini, Silicon Biosystems, Bologna, Italy) and the cells were loaded onto the DEPArray NxT cartridge following the protocol instructions. CellBrowser™ (Menarini, Silicon Biosystems, Bologna, Italy) analysis software, integrated into the DEPArray™ system, allows the user to view and select cells from the particle database according to multiple criteria, based on qualitative and quantitative marker evaluation and cell morphology. This software enables the user to create populations and sub-populations of cells using analysis tools such as scatter plots, histograms, and image panels. Cells become un-routable based on their positions; when these are out of the cage, it is no longer possible to move them and therefore complete the recovery. First of all, we excluded clusters of two or three cells, clumps, and spurious events and focused only on single cells with the desired fluorescence, analyzing only the “centered” DAPI cells in the cage. The single cells were selected manually based on fluorescence labeling and morphology. About 20 different single cells were recovered for each tumor patient and volume reduction was performed with a VRNxT-Volume Reduction Instrument (Menarini, Silicon Biosystems, Bologna, Italy) according to the instruction manual. The isolated cells were stored at −20 °C until later downstream analysis.

### 2.5. Immunofluorescence of GB Tissues

Formalin-fixed, paraffin-embedded (FFPE) tissue blocks, obtained from our GB samples, were cut into 2–4 μm thick sections. Antigen unmasking was achieved with Epitote Retrieval Solution (pH = 8) (Leica Microsystems, Wetzlar, Germany) in a microwave. GFAP monoclonal (ASTRO6) (MA5-12023, Thermofisher Scientific, Waltham, MA, USA) and IBA1 polyclonal (091-19741, Wako Chemicals, Neuss, Germany) primary antibodies were then applied at dilutions of 1:100 and 1:1000, respectively, overnight at 4 °C. The goat anti-rabbit Alexa Fluor 488 (Thermofisher Scientific, Waltham, MA, USA ) and goat anti-mouse Alexa Fluor 568 (Thermofisher Scientific, Waltham, MA, USA ) were diluted 1:500 and incubated for 1 h. Cells were counterstained with DAPI (Sigma Aldrich, St. Louis, MO, USA) and visualized using an Olympus Fluoview 3000 confocal microscope at a magnification of 60×.

### 2.6. DNA Extraction from Fresh Tissues

Genomic DNA was extracted directly from up to 50 mg of fresh tissue of GB01, GB02, and GB03 using the Maxwell^®^ 16 Instrument with the Maxwell^®^ 16 Tissue DNA Purification Kit (Promega, Madison, WI, USA). DNA concentration was determined using the Qubit Fluorometer (Life Technologies, Carlsbad, CA, USA) and the quality was assessed using the Agilent 2200 Tapestation (Agilent Technologies, Santa Clara, CA, USA) system.

### 2.7. Ampli1™ Whole Genome Amplification and Low Pass Analysis

Whole-genome amplification on all recovered single cells was performed using the Ampli1™ WGA Kit version 02 (Menarini, Silicon Biosystems, Bologna, Italy) following the manufacturer’s instructions. The same procedure was adjusted for the DNA obtained from fresh tissues starting from 1 µL of 1 ng/µL. Afterward, the WGA product was cleaned up with SPRIselect Beads (Beckman Coulter, Brea, CA, USA) and sequencing-ready libraries were prepared with an Ampli ™ LowPass Kit (Menarini, Silicon Biosystems, Bologna, Italy) to detect chromosomal aneuploidies and copy number aberrations (CNAs) with a low sequencing depth. To sequence our libraries, we used an Ion 520/530-OT2 kit (Ion Torrent, Life Technologies, Grand Island, NY, USA) with the Ion 530 Chip (Ion Torrent, Life Technologies, Grand Island, NY, USA). The runs were conducted on the Ion S5 system (Ion Torrent, Life Technologies, Grand Island, NY, USA).

### 2.8. CNA Calling

The data obtained from low-pass whole genome sequencing were processed with the IchorCNA tool [17]. The CNA segmented number profiles obtained from IchorCNA were processed with the CNApp tool [18] with default cutoffs.

## 3. Results

### 3.1. Isolation of Single-Cells from GB Fresh Tissues with DEPArray^TM^ NxT

Three GB fresh tissues obtained from the Unit of Neurosurgery of Livorno Civil Hospital were analyzed with DEPArray^TM^ NxT, the overview of the procedure is shown in Figure 1 in which H&E images for each tumor tissue are also present. After DEPArray NxT Cartridge loading, we selected the routable cells using the CellBrowser™ analysis software. In detail, for GB01, 2880 routable cells, for GB02, 17,378 routable cells, and for GB03, 4788 routable cells, were observed. After that, we performed the exclusion of cell clusters obtaining single and routable cells: 2654, 9535, and 4278 cells respectively for GB01, GB02, and GB03.

#### Cell Populations in GB01, GB02, and GB03

We chose four different conventional markers to identify the four most representative subpopulations of GB (astrocytes, microglia, endothelial cells, and stem cells): GFAP, IBA1, CD105, and CD133. We decided to call them: astrocyte-like, microglia-like, endothelial-like, and stem-like cells because of their similarity to these particular cells. Moreover, we found some cells with double fluorescence staining.

An example of the main populations is shown in Figure 2. In Figure 3, percentages of the main populations found in the three samples are summarized, while in the Appendix A, double fluorescence stained cells and unlabeled cells are shown. We recovered both single cells and groups of a maximum of five single cells with the same characteristics. The recovered cells for the three samples are summarized in Figure 4. In particular, for GB01 we selected 20 cells: 3 single astrocytes-like, 3 groups of astrocytes-like, 4 microglia-like single cells, 2 groups of microglia-like cells, 1 group of endothelial-like cells, 1 single stem-like cell, 2 single astrocytes/microglia-like cells (positive for both GFAP and IBA1), and 3 single cells and 1 group of single cells without labeling (positive to Hoechst 33342 only). For GB02, 26 cells were recovered: 6 single astrocytes-like, 5 microglia-like single cells, 5 single endothelial-like cells, 3 groups of endothelial-like cells, 5 single stem/endothelial-like cells (positive both for CD133 and CD105), 1 group of stem/endothelial-like cells (positive both for CD133 and CD105), and 1 single cell without labeling (Hoechst 33342 signal only). Finally, for GB03, 17 cells were selected: 6 single astrocytes-like, 5 single microglia-like cells, and 6 single endothelial-like cells.

### 3.2. Copy Number Aberrations (CNAs) Analysis

#### 3.2.1. GB Bulk Tissues

Cellular genomic profiling was performed on the selected cells using the Ampli1™ LowPass kit to identify genome-wide CNAs at the single-cell level and to obtain information on ITH. The same analysis was also carried out on the DNA obtained from fresh tumor tissues (GB01, GB02, and GB03), to compare the bulk molecular profile to the one derived from single cells. In Figure 5, the CNA pattern of the fresh GB tissues is shown: as expected, each sample has a different CNA configuration due to GB ITH. However, all three samples presented chromosome 10q deletion, and GB01 and GB02 also presented chromosome 7 amplification, which represent typical GB alterations. Consequently, for each sample, tumors in bulk and single-cell CNAs were compared.

#### 3.2.2. GB Single Cells

The summarized results obtained from CNA analysis on single cells are described in Table 2, Table 3 and Table 4. In GB01, we found a group of wild type endothelial-like cells; of these, there were six microglia-like cells (four single cells and two groups of cells), two were wild type (one single cell and one group of cells), one cell showed a chr 19 deletion only, and the other cells showed different alterations, sharing a chr 10 deletion, and chr 7, 9q, and 17q amplifications; six astrocytes-like (three single cells and three groups of cells) were altered, sharing a chr 10 deletion, and chr 7, 9q, and 17q amplifications; one stem-like cell with a chr 1p and 10 deletion and chr 7, 9, 17q, and 19q amplifications. In GB01, moreover, two cells with double staining (GFAP and IBA1) were found with the same alterations, chr 1p and 10 deletions and chr 7, 9, 17q, and 19q amplifications. Finally, three out of four unstained cells (three single and one group of cells) were wild type and one exhibited chr 1p, 10, and 17p deletion and chr 7, 9q, 17q, and 19q amplifications. In GB02, we found eight endothelial-like cells (five single and three groups of cells), two were wild type and the others carried a chr 19 deletion except for only one having chr 9p, 10, 13q, 14q, and 22q deletions. Then, of six single astrocytes-like cells, one was wild type, one had a chr 19 deletion, and the others shared chr 9p, 10, 13q, 14q, and 22q deletions. Indeed, of five single microglia-like cells, two were wild type and three had a chr 19 deletion. In GB02, we selected six double staining cells (CD133 and CD105 positive), of these, four were wild type and the others shared chr 10, 13q, 14q, and 22q deletions. Finally, one unstained cell was wild type. GB03 counted six single endothelial-like cells, four of which were wild type and the other two presented different alterations sharing in particular chr 9p, 10, and 22q deletion and chr 7, 9q, and 20 amplifications. Five single microglia-like cells were all wild type. Finally, six single astrocytes-like were selected, one was wild type while the other cells showed all the same alterations: chr 9p, 10, and 22q deletions and chr 7, 9q, and 20 amplifications.

#### 3.2.3. Comparison between Bulk Tissues and Single Cells

In Figure 6, the comparison between bulk fresh tumor CNAs and single cell CNAs obtained with the CNApp tool is shown. Cells with CNAs have similar alterations to those found in the bulk tissues and also show additional alterations. The molecular alteration profiles in single cells are more strongly highlighted, as in bulk tumors many alterations may be hidden since many cells are analyzed at the same time.

### 3.3. Double Staining Cells Immunofluorescence

To confirm the presence of double staining cells in our tissues, we performed immunofluorescence with anti-GFAP (red) and anti-IBA1 (green) on our tissues’ slides. We observed some cells with these characteristics in GB01 tissue slides, as shown in Figure 7.

## 4. Discussion

Despite the new therapies developed in the last few years, GB still remains an incurable and devastating disease [19]. The adjective “*multiforme*”, often used to define GB, was coined in 1926 by Percival Bailey and Harvey Cushing [20] to describe the various appearances of necrosis, cysts, and hemorrhage. As a matter of fact, this definition also fits from a molecular point of view to explain the high degree of heterogeneity in GB. The poor prognosis of GB patients is mainly associated with ITH, which represents the presence in the tumor mass of multiple sub-clones, each characterized by different molecular and genomic alterations [21]. The sub-clones’ alterations are certainly masked during bulk tumor analysis [22]. There are several approaches to assess the degree of ITH, such as flow cytometry or more innovative methods such as single-cell sequencing and DEPArray analysis. These are certainly three technologies used to decipher ITH, but none can replace the others; rather, they aim to be complementary. Recently, in some single-cell sequencing studies, to investigate the ITH, CNAs investigations were conducted instead of the identification of individual mutations with a gain in sensitivity and accuracy [4,15,23]. Regarding these different techniques, single-cell RNA seq is mainly a discovery analysis: recent single-cell transcriptome studies in GB have made it possible to identify tumor cell populations and to highlight tumor plasticity and hierarchy [24,25]. DEPArray analysis, instead, allows us to select, isolate, and analyze specific cells or groups of cells providing a higher level of precision and accuracy in cell selection than traditional flow cytometry, with a high transfer efficiency and unprecedented purity for molecular analysis. In this work, we decided to focus our attention on some of the most representative GB populations: astrocytes, microglia, stem cells, and endothelial cells. We have assumed that we have isolated the above-mentioned cells, based on the positivity of the chosen markers. However, as only one marker is used *per* cell, we cannot be sure that we have exactly the hypothesized cell, so we use the suffix *“–like”* to describe the cells isolated. Through selection and isolation with DEPArray, we investigated the molecular alterations of the isolated cells by comparing them with whole tumor tissue, in terms of CNAs. Astrocytes are star-shaped cells of the brain with different active roles in both healthy people and in brain pathological conditions [26]. For example, they regulate neural signaling and give support in blood-brain barrier (BBB) formation [26]. Regarding GB tumorigenesis, a much-debated topic concerns the cell-of-origin in the cancer stem cell (CSC) or hierarchical hypothesis: GB stem cells (GSCs) or glioma initiating cells seem to be responsible for tumor formation [3]. They are a small population of stem cells characterized by self-renewal and differentiation properties [27]. GSCs are involved in tumor growth, invasion, and recurrence development [28]. Based on this theory, GSCs can arise from neural stem cells [29] but also from already differentiated astrocytes transformed through genetic and epigenetic mutations [30,31]. Therefore, based on this hypothesis, the cell population initiating GB is composed of a mixture of cells including astrocytes and stem cells. In our work, most of the astrocytes-like cells in all three tumors, were altered with a CNA pattern identical to the bulk tumor. In some cells, more alterations were observed than in the bulk, in support of the concept of the higher sensitivity and accuracy of the single-cell analysis approach. Indeed, the only stem-like cell collected in GB02 showed a CNA pattern typical of a transformed tumor cell. This suggests that the cumulative acquisition of mutations in the stem cells can be responsible for invasive cancer generation.

In the brain, microglial cells, a specialized population of macrophage-like cells, represent resident innate immune cells and are involved in many crucial physiological processes [32]. Microglia have been ignored for a long time but by now it is common knowledge that these cells are an integral part of the tumor, constituting approximately 30% of tumor mass [33] and participating in tumor progression and anti-cancer treatment resistance [34]. Indeed, microglial cells have a key role in many brain diseases [35]. From our results, we observed some microglia-like cells with normal chromosomes sets, as we expected, but we also found some cells presenting CNAs, indicating that within the tumor there are also microglia cells with potential tumoral behavior. From a transcriptional point of view, some alterations have been described in GB microglia [36]. In 2020, Maas and colleagues defined a particular type of transformed microglial cells. In this context, tumoral GB cells hijack microglial gene expression to enhance tumor proliferation, suppressing the immune response [37].

Endothelial cells (ECs) represent the principal components of the BBB [38]. Different brain pathologies, including GB, show molecular alterations of ECs [39]. In GB, vessels are necessary for cancer cell spreading and it has been demonstrated that ECs regulate tumor invasion through crosstalk with GB cells [40]. Our results illustrate the presence of wild type endothelial-like cells also carrying CNAs, confirming that the tumor mass can contain tumor-ECs (also defined tumor-associated ECs) as has been highlighted in some recent publications [41,42,43]. In these papers, the tumor-associated ECs showed different phenotypic and functional characteristics concerning normal ECs. Moreover, the relationship between ECs and GB tumor cells was demonstrated in two recent studies, in particular it was observed that tumor-derived ECs and GB stem cells shared the same genomic mutations and that CD144 and VDGFR2 genes are expressed by the emerging endothelium [44,45].

Moreover, in our study, we observed and then recovered some cells with a double signal of labeling: astrocytes/microglia-like cells in GB01 and stem/endothelial-like cells in GB02. Indeed, in the literature, the detection of dual positive cells has been reported in experiments using our same technology, especially in the circulating tumor cell studies [46]. The presence of these double stained cells, in particular GFAP+/IBA1+, was also confirmed by immunofluorescence experiments (Appendix A) to strengthen our findings. Fais et al. in 2007 introduced the concept of cannibalism as an exclusive property of malignant tumor cells [47]. Moreover, Coopman et al. assumed that phagocytosis is the mechanism used by invasive tumor cells to allow migration into the surrounding tissues [48]. In this regard, in malignant gliomas, phagocytic tumor cells were detected, particularly in GB [49,50]. A different hypothesis could be the cell fusion formation, for example, Huysentruyt et al. observed fusion between macrophages and tumor cells [51].

A further aspect that emerged from our results is the detection both in GB01 and in GB02 of some unstained cells with CNAs. We observed, in fact, that not all the astrocytes are positive for GFAP and it has also been demonstrated in the literature that GFAP is not an astrocytes-exclusive marker, as GFAP expression in GB varies significantly [52].

The use of CNAs as a method of evaluating tumor cells is more popular lately. The CNA burden is assessed in different tumors, such as in prostate cancer [53], meaning as the analysis of the variable amounts of amplifications or deletions in different patients. In particular, Hieronymus et al. [53] observed that patients with a high CNA burden showed a greater risk of relapse after treatment. For this reason, CNA analysis can also be considered as a useful marker. Therefore, the tumor CNA burden, rather than individual CNAs, can be associated with cancer outcomes. Recently, CNA analysis has been evaluated as more advantageous than mutational analysis for diagnostic reasons in particular in association with survival [54]: CNAs and miRNA analysis had a better performance than mutational data for poorly predicted survival. In addition, in melanoma, Roh et al. demonstrated that the association of CNAs and the mutational burden can be very useful for prognosis and the response to therapy [55].

To the best of our knowledge, this is the first time that our approach is used to partition a GB tumor tissue into its cellular components and provide its molecular profile. Single-cell CNA analysis has the potential to yield new insights into the molecular dynamics of cellular populations. Measuring single-cell genome alterations in tissues and cell populations will greatly advance the clonal decomposition of malignant tissues, resolving rare cell population genotypes and identifying DNA amplification and the deletion states of individual cells, which are difficult to establish when cellular information is destroyed in bulk sequencing. A novel feature of our approach is also the capture, by brightfield and immunofluorescence imaging, of the morphologic features of cells, permitting analytical integration with genomic properties.

## 5. Conclusions

In conclusion, in this work, we were able to isolate single cells from fresh GB tissues based on markers that assigned them to the four cell subpopulations: astrocytes, microglia, endothelial cells, and stem cells. CNA analysis allows us to distinguish the tumor cells inside the tumor microenvironment. This is a preliminary work based on an innovative technique, single-cell CNA analysis with DEPArray, to select single tumor cells and study their molecular alterations in depth. This new type of experimental approach is proposed as a complementary procedure to conventional methodologies and provides a baseline for further analyses that aim to explore in depth the different subpopulations in the GB microenvironment. Moreover, the single-cell approach allows a very sensitive analysis rather than bulk analysis, obtaining molecular profiles more accurately. In such an aggressive and lethal tumor, any kind of information is crucial and can be useful to better understand the mechanisms underlying the development of the tumor and its propagation.

## Figures and Tables

**Figure 1 cells-11-01127-f001:**
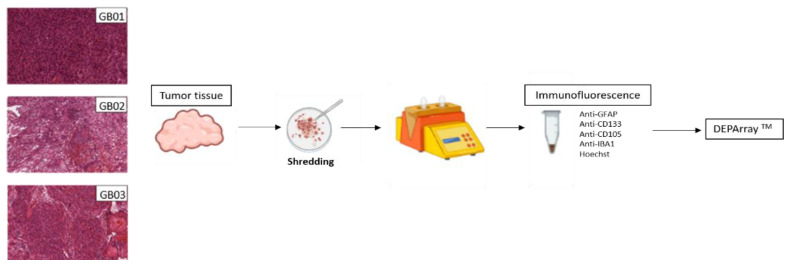
Histological images of GB01, GB02, and GB03. Experimental design starting from tumor shredding to DEPArray analysis.

**Figure 2 cells-11-01127-f002:**
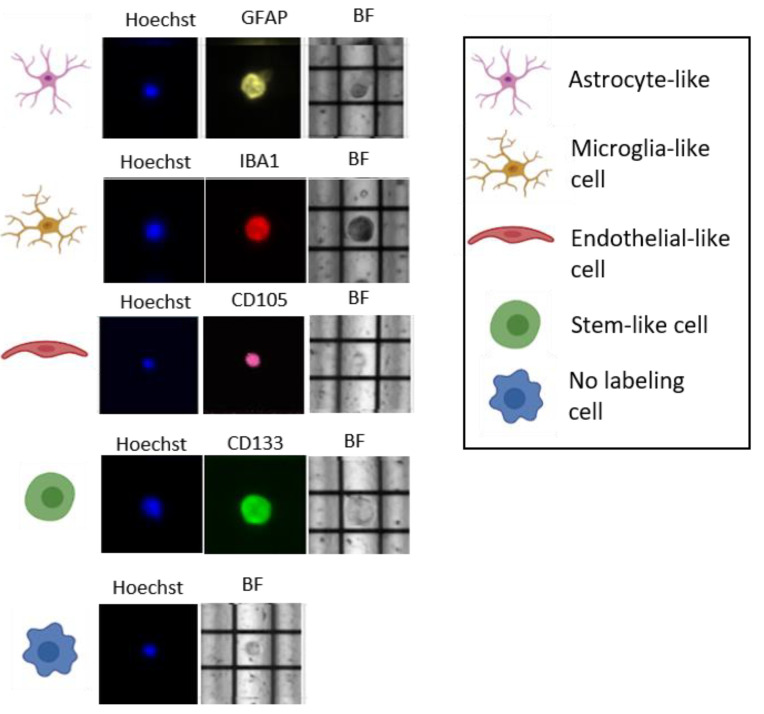
Example of DEPArray images of single cells belonging to the main GB populations, stained in yellow with GFAP (astrocytes-like), in red with IBA1 (microglia-like), in purple with CD105 (endothelial-like cells), in green with CD133 (stem-like cells) and in blue with Hoechst. BF: Brightfield.

**Figure 3 cells-11-01127-f003:**
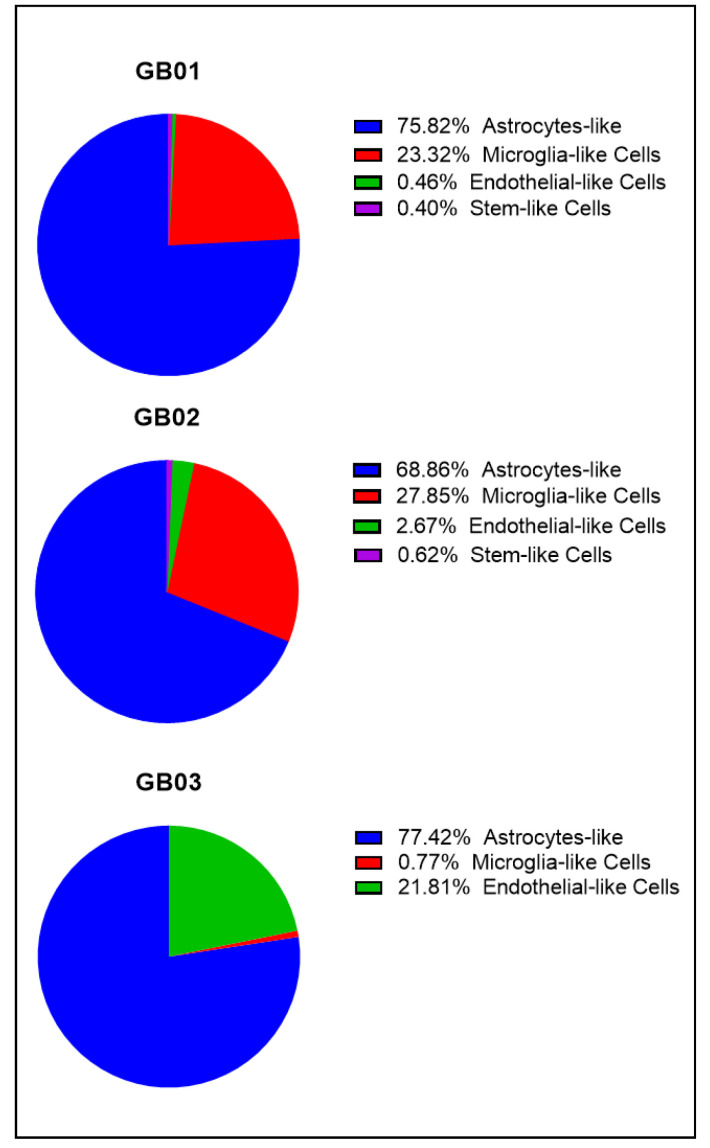
Pie charts of the percentages of the main cell type populations found in GB01, GB02, and GB03.

**Figure 4 cells-11-01127-f004:**
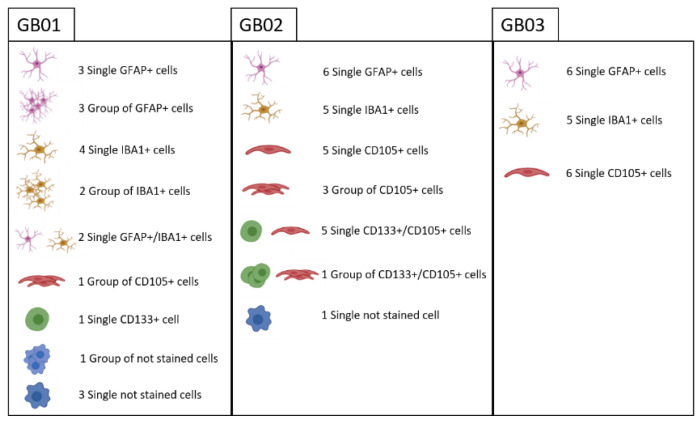
Summary of all the recovered cells after DEPArray analysis from GB01, GB02, and GB03. Astrocytes-like, microglia-like, endothelial-like, and stem-like cells were collected. Double staining cells and only Hoechst positive cells are shown.

**Figure 5 cells-11-01127-f005:**
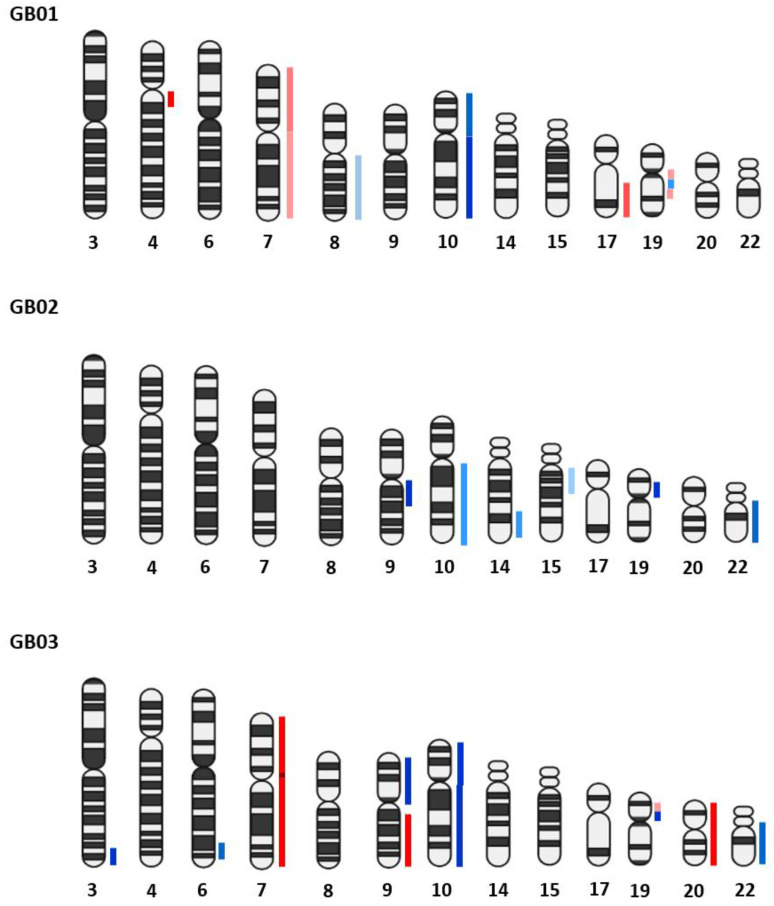
CNA pattern of the fresh GB tissues in bulk. The chromosomal amplifications are shown in red, and in blue the deletions. The intensity of the red and blue color components correlates with the gain and loss values based on the results obtained from the CNApp tool.

**Figure 6 cells-11-01127-f006:**
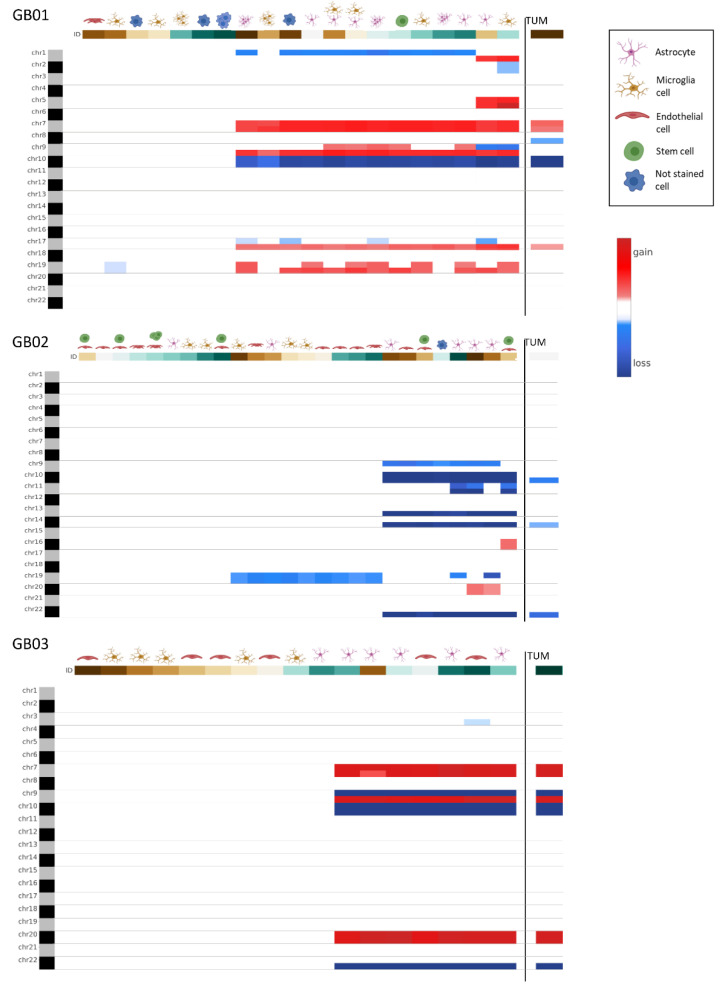
Genome-wide chromosome arm CNA profile heatmap for GB01, GB02, and GB03. For each sample, the CNA profile of the single cells collected is shown and on the right the CNA profile of the tumor tissue in bulk.

**Figure 7 cells-11-01127-f007:**
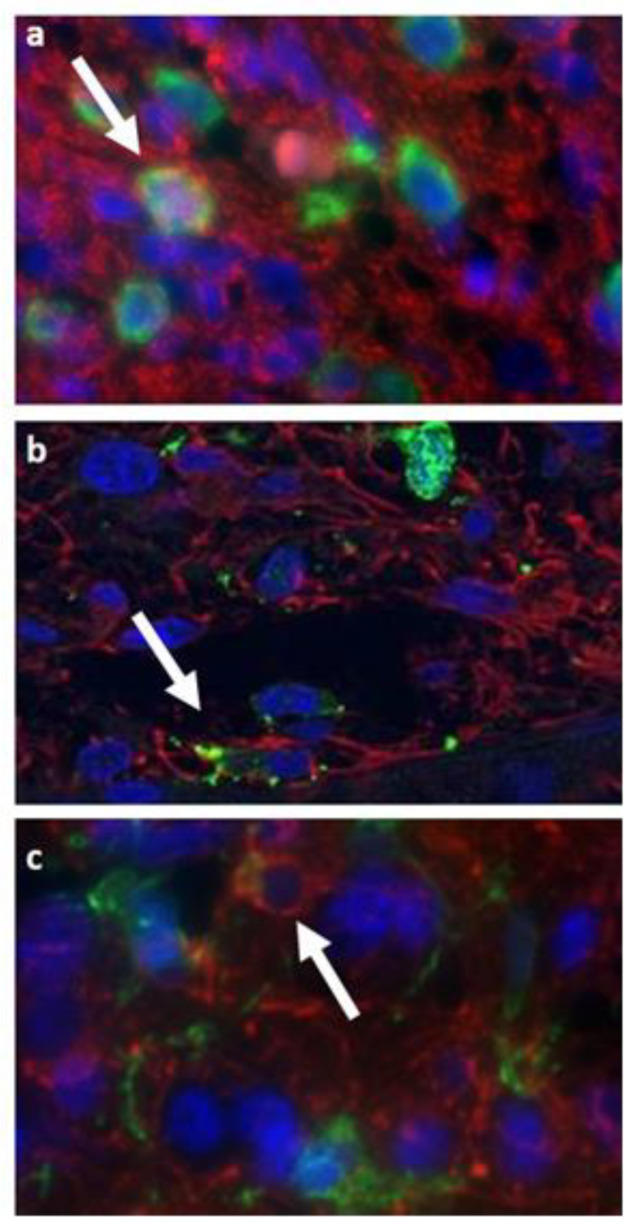
Immunofluorescence assays of GB01 tissue: (**a**–**c**) three different portions of GB01 tissue with GFAP (red) and IBA1 (green) markers shown. Arrows indicate cells with the double-positive signal.

**Table 1 cells-11-01127-t001:** Patient clinical, demographic, pathological, and therapeutical data.

Cases	Age	Sex	Primary or Recurrence	Brain Location	IDH1/IDH2	Pathology Report	Therapy Administered
GB01	30	M	Primary	parietal lobe	WT	Glioblastoma (Grade IV WHO) (GFAP+, MKI67-20%)	Levetiracetam, Soldesam, Lansoprazole
GB02	47	M	Primary	right temporal lobe	WT	Glioblastoma (Grade IV WHO) (GFAP+, MKI67-30%)	Levetiracetam, Dexamethasone, Omeprazole
GB03	65	M	Primary	right frontal lobe	WT	Glioblastoma (Grade IV WHO) (GFAP+, MKI67-20%)	Levetiracetam, Lansoprazole, Dexamethasone (Mannitol pre-op)

**Table 2 cells-11-01127-t002:** CNAs results obtained after CNApp processing for single cells and groups of single cells collected in the GB01 sample.

GB01	
**Single Cells Collected**	**CNA**
Group of endothelial-like cells	WT
Microglia-like cell	19-
Microglia-like cell	WT
Microglia-like cell	1p-, 7+, 9q+, 10-, 17q+, 19+
Microglia-like cell	1q+, 2-, 5+, 7+, 9p-, 9q+, 10-, 17q+, 19+
Group of microglia-like cells	WT
Group of microglia-like cells	7+, 9q+, 10-, 17q+
Astrocyte-like	1p-, 7+, 9q+, 10-, 17q+, 19+
Astrocyte-like	1p-, 7+, 9q+, 10-, 17q+, 19+
Astrocyte-like	1q+, 5+, 7+, 9p-, 9q+, 10-, 17p-, 17q+, 19q+
Group of astrocytes-like	1p-, 7+, 9q+, 10-, 17p-, 17q+, 19+
Group of astrocytes-like	1p-, 7+, 9+, 10-, 17p-, 17q+, 19+
Group of astrocytes-like	1p-, 7+, 9q+, 10-, 17q+
Stem-like cell	1p-, 7+, 9+, 10-, 17q+, 19q+
Astrocyte/microglia-like cell	1p-, 7+, 9+, 10-, 17q+, 19q+
Astrocyte/microglia-like cell	1p-, 7+, 9+, 10-, 17q+, 19+
Not stained cell	WT
Not stained cell	WT
Not stained cell	1p-, 7+, 9q+, 10-, 17p-, 17q+, 19q+
Group of not stained cells	WT

**Table 3 cells-11-01127-t003:** CNAs results obtained after CNApp processing for single cells and groups of single cells collected in the GB02 sample.

GB02	
**Single Cells Collected**	**CNA**
Endothelial-like cell	WT
Endothelial-like cell	19-
Endothelial-like cell	19-
Endothelial-like cell	19-
Endothelial-like cell	9p-, 10-, 13q-, 14q-, 22q-
Group of endothelial-like cells	WT
Group of endothelial-like cells	19-
Group of endothelial-like cells	19-
Astrocyte-like	WT
Astrocyte-like	19-
Astrocyte-like	9p-, 10-, 13q-, 14q-, 22q-
Astrocyte-like	9p-, 10-, 11- 13q-, 14q-, 19p-, 22q-
Astrocyte-like	9p-, 10-, 11- 13q-, 14q-, 20+, 22q-
Astrocyte-like	9p-, 10-, 13q-, 14q-, 19p-, 20+ 22q-
Microglia-like cell	WT
Microglia-like cell	WT
Microglia-like cell	19-
Microglia-like cell	19-
Microglia-like cell	19-
Endothelial/stem-like cell	WT
Endothelial/stem-like cell	WT
Endothelial/stem-like cell	WT
Endothelial/stem-like cell	9p-, 10-, 13q-, 14q-, 22q-
Endothelial/stem-like cell	10-, 11-, 13q-, 14q-, 16+, 22q-
Group of endothelial/stem-like cells	WT
Not stained cell	9p-, 10-, 13q-, 14q-, 22q-

**Table 4 cells-11-01127-t004:** CNAs results obtained after CNApp processing for single cells and groups of single cells collected in the GB03 sample.

GB03	
**Single Cells Collected**	**CNA**
Endothelial-like cell	WT
Endothelial-like cell	WT
Endothelial-like cell	WT
Endothelial-like cell	WT
Endothelial-like cell	7+, 9p-, 9q+, 10-, 20+, 22q-
Endothelial-like cell	3q-, 7+, 9p-, 9q+, 10-, 20+, 22q-
Microglia-like cell	WT
Microglia-like cell	WT
Microglia-like cell	WT
Microglia-like cell	WT
Microglia-like cell	WT
Astrocyte-like	WT
Astrocyte-like	7+, 9p-, 9q+, 10-, 20+, 22q-
Astrocyte-like	7+, 9p-, 9q+, 10-, 20+, 22q-
Astrocyte-like	7+, 9p-, 9q+, 10-, 20+, 22q-
Astrocyte-like	7+, 9p-, 9q+, 10-, 20+, 22q-
Astrocyte-like	7+, 9p-, 9q+, 10-, 20+, 22q-

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
