# Peer review of "Single-Cell Molecular Characterization to Partition the Human Glioblastoma Tumor Microenvironment Genetic Background"

_cells, 2022, doi:10.3390/cells11071127_

Round 1

Reviewer 1 Report

The manuscript entitled Single-cell molecular characterization to partition the human glioblastoma tumor microenvironment genetic background addresses a topical issue in glioblastoma research area, yet it needs some improvements in order to be published.

In the material and method chapter, the amount (grams) of tissue started must be mentioned in each analysis.

In the results chapter a problem is represented by the small number of cells that were marked/labelled in the case of samples GB01 and GB03, data presented in Figure 1S. Subsequent analysis between bulk cells and isolated cells may be affected by this tehnical problem.

Also, for a better understanding of the obtained results, I suggest renaming the subchapters from the results with the main scientific conclusion of the respective subpoint.

Line 224, GB03 should be mentioned instead of GB02, according to figure 5.

The subchapter Double staining cells immunofluorescence is incorrectly numbered 3.2 instead of 3.3 and the images that support this fact must be removed from additional data (Figure 2S) and inserted in the article main body. Additional experiments should support this subchapter.

Line 348, references should be like (49,50), not like (49) (50).

A separate chapter of conclusions should be mentioned, with emphasis on comments on importance, validity and generality of conclusions.

Considering the above mentioned suggestions, we recommend the publication of the manuscript after major revisions are made; still the final decision belongs to Editor-in-chief.

Author Response

We would like to thank the reviewer for his/her thoughtful comments. We will answer point by point to the reviewer’s comments:

  • We added in the Materials and Methods section the amount of tissues used for the analysis
  • We apologize for the misunderstanding, probably we did not explain the concept very well in the text. Figure 1S only concerns double positive cells which were in fact very few, in GB01 and GB03. We have shown them only in the Supplementary data for this reason. We did not want to explore in deep on these cells, but only show them for completeness of the work.
  • We added subchapters in the Results section, as the reviewer suggests.
  • We corrected the sentence.
  • We corrected the error and added the figure in the paper. We would just like to explain that the immunofluorescence assay was performed only to confirm the presence in our tissues of the double positive cells, as we found with DEPArray analysis.
  • We merged the references.
  • We added a separate paragraph with the conclusions, as suggested by the reviewer.

We would like to thank the referee again for taking the time to review our manuscript.

Reviewer 2 Report

Minor comments

The article is well written. It is rather descriptive, however, the general conclusions are in accordance with the results obtained.

Each single cell obtained by DEPArray must be obtained by at least two (or three) markers to define whether it is a microglia, astrocyte or stem cell. It is difficult to classify populations by the expression of only one single stain marker and morphology. For example, it has been well described that there are stem cell subtypes that do not express CD133 and instead express both CD44 and CD105, which in this article could be erroneously classified as endothelial. Furthermore, CD133 and GFAP expression has also been described in both non-tumor astrocytes and in tumor cells such as U87.

Although it is well explained in supplementary figures 1 and 2, authors textually state: "Afterward, we identified four main populations (astrocytes, microglia cells, endothelial cells and stem cells) and several cells with double fluorescence staining. staining." Having only single and not double staining is not an adequate criterion to classify each cell.

General sugestion: I suggest rewriting the paper and only show the CNA profile by markers but not by cell type.

Author Response

We would like to thank the reviewer for his/her thoughtful comments.

We agree with the reviewer that only one marker is not sufficient to define for sure the specific cell. Nevertheless, the goal of our study is to discriminate several conventional populations within the GB and to study their chromosomal aberrations through CNA analysis. In this type of approach, we have chosen the four most commonly used generic markers trying to mark as many cell populations as possible. DEPArray instrument allows to analyze the samples exploiting 5 different fluorescence channels, so the markers must be chosen depending on the investigation to be pursued. In this case, we, therefore, used only one marker for each type of population. Anyway, we think that it is important to mention the cell type in order to understand the possibility of identifying different populations in a tumour tissue like GB that is so heterogeneous. This work represents a starting point, further analyses could be performed, for example several runs on the same sample with more markers to select exactly the type of cells that are being observed. However, since we understand the reviewer’s point of view, we have decided to add the suffix -like in the paper with reference to the cells we isolated, so as to be sure that no false statements are made. We hope this was the right decision and that the reviewer understand our point.

We would like to thank the referee again for taking the time to review our manuscript.

Round 2

Reviewer 1 Report

For me the changes are fine.